# Prevalence of the Preschool Teachers' Physical Activity Level: The Case of the Republic of Croatia

**Vilko Petrić [1]**, **Bruna Francetić [2,*]** and **Lidija Vujičić [1]**

1   Faculty of Teacher Education, University of Rijeka, Sveučilišna avenija 6, 51000 Rijeka, Croatia; vilko.petric@uniri.hr (V.P.); lidija.vujicic@uniri.hr (L.V.)
2   Kindergarten Rijeka, Veslarska 5, 51000 Rijeka, Croatia
*   Correspondence: brunafrancetic1@gmail.com

**Abstract:** The aim of this article was to determine the prevalence of the Republic of Croatia preschool teachers' physical activity level and its correlation to chronological age, years of service, degree of education, and place of work. The research was conducted on a sample of 159 preschool teachers from different parts of the Republic of Croatia. The Croatian version of the standardised version of the International Physical Activity Questionnaire (IPAQ) was used. The basic descriptive parameters were calculated, while the correlation between certain variables was determined by the Spearman correlation coefficient and linear regression analysis. Results have shown that almost 80% of preschool teachers do not meet the criteria for the recommended levels of physical activity. They are most physically active at work but are the least active during transportation and leisure time. A statistically significant correlation was determined between physical activity and age, years of service, and place of work. Younger preschool teachers are significantly more physically active in their leisure time than older ones, as are those with fewer years of service. Preschool teachers who work in a bigger city are more physically active at work, whereas those from smaller cities are more active doing their household chores. Furthermore, the independent variables (Age, Years of service, Place of work, Degree of education) in combination significantly affect the level of physical activity of preschool teachers in their leisure time and the overall level of physical activity. Preschool teachers should be models for children, and if they are not physically active themselves, they will not be sufficiently motivated to stimulate the same in children, i.e., the habit of leading an active and healthy life from an early age.

**Keywords:** physical activity; preschool teacher; chronological age; years of service; degree of education; place of work

## 1. Introduction

Insufficient physical activity is becoming a significant risk factor for developing the most common diseases of modern times and is in fourth place for mortality risks in the world [1]. The recommendations given by competent institutions for adults state that to achieve health benefits, they should be involved in moderate-intensity physical activities for at least 150 min a week [2]. Research on this topic in the last few years has been regularly conducted to detect risk factors that could help create strategies intended to increase physical activity, especially in segments where it is the lowest [3]. This research represents an important indicator of health and lifestyle in certain groups of people, whereas the obtained results give an insight into one's health condition, use of health protection, and health determinants. The research analysis indicates areas that need interventions and motivation to acquire healthy everyday life habits to improve health and diminish the death rate caused by chronic non-infectious diseases [4]. Children are the fundaments of society and healthy life habits are acquired from an early age, so the primary focus is put on early and preschool age and early education institutions [5]. Children who are

a part of the educational system spend almost the whole of their active daily routine in educational institutions with peers and adults while an intimate link is built with their preschool teacher who is often mentioned along with their narrower family members [6].

The latest Eurostat research [7] has found that a fourth of the European population (27%) does exercises for three hours a week, 17% do exercises from three to five hours, while 28% do exercises for more than five hours a week. Results have also confirmed that the number of inactive people keeps increasing every year. As many as 59% of the European Union citizens exercise never or rarely, with 41% doing so at least once a week. Citizens coming from north European countries are significantly more active than citizens from the south or east. Sweden is at the top of the list, with 70% of participants stating they exercise at least once a week, whereas Bulgaria is at the bottom of the list with 78% of participants not engaging in physical activities during the week. It is important to mention that Croatia is also at the bottom as the country whose population is the least active. In the Republic of Croatia, 56% of citizens do not exercise or do sports, while 20% do so rarely. Only 5% of the citizens exercise regularly during the week. According to employment criteria, only 29.67% of male employees and 32.75% of female employees have the recommended physical activity level [8].

Understanding the importance of movement for the overall maturity of a person, the preschool teacher enables the child to feel out their boundaries and investigate their own world in the educational process. The preschool teacher becomes a model and is a direct influence on the child's ability to put challenges into practice. It becomes evident that if preschool teachers want to achieve this, they must understand the benefit brought by movement and be aware of the importance of its implementation into the educational process. Given that a preschool teacher has a strong role model influence on early and preschool-aged children, their love for healthy and active living is relevant. It is known now that if preschool teachers are not physically active, they will not be sufficiently motivated to encourage children to be active and develop love towards an active and healthy lifestyle from an early childhood age [9]. Physical activity is a very significant segment of the preschool teachers' profession since one of the primary tasks of educational work should be to create a culture of movement among early and preschool-aged children. Therefore, with their pattern of behaviour, preschool teachers should incite children to live in movement and be a model for its achievement.

Consequently, the aim of this research was to determine the prevalence of the Republic of Croatia preschool teachers' physical activity level and its correlation to chronological age, years of service, degree of education, and place of work.

## 2. Materials and Methods

### 2.1. Research Participants

The research was done on a convenient sample of 159 preschool teachers from different educational institutions in the territory of the Republic of Croatia. The division of the participants was done according to the size of their place of work. The criteria of smaller vs. bigger cities were determined on the number of citizens being 15,000 (less than that was considered a smaller city, more than that, a bigger one). Of all the participants, 39% worked in bigger cities, and 61% worked in smaller cities, by the above-mentioned criteria.

Most of the participants were females (97.5%, N = 155), and their average age was 34 (min = 22, max = 64). The number of years of service ranged from half a year to 42 years, while the average value was 9. A total of 42% of participants completed a graduate university study, 30% an undergraduate university study, and 27% of them marked college education or undergraduate professional study as their achieved degree of education. Only 1% marked 'Other' and explained it with a postgraduate university study.

### 2.2. Description of the Instrument and Variables

For the needs of this research, the Croatian version of the International Physical Activity Questionnaire was used [10].

The questions of the questionnaire are divided into five categories: 1. Physical activity at work, 2. Physical activity during transportation, 3. House chores, maintaining the house,

and care of the family, 4. Recreational activities, sports, and leisure-time physical activities, and 5. Time spent sitting. The items are interconnected so that the participant responds to the amount of time spent on extremely vigorous and moderate physical activities in the last seven days. Each question is answered on a scale from 0 to 7 days and then in minutes from 0 to 240; intervals of 30 min are in the form of multiple-choice questions.

The sociodemographic data about the preschool teachers participating in the research represent the research's criterion variables: gender, age, years of service as a preschool teacher, place of work, and the highest achieved degree of education. For questions regarding age, years of service as a preschool teacher, and place of work, open-ended questions are used. The question about the participants' gender is dichotomous, with number 1 describing males and number 2 females. For the variable regarding the highest degree of education, multiple-choice questions are used with answers ranging from 1—secondary school level, 2—college or undergraduate professional study, 3—undergraduate university study, 4—graduate university study, and 5—other.

### 2.3. Description and Research Protocol

The research was conducted in April 2021 via the internet application Google forms. Data were collected in 19 days. The link to participate in the research was sent by electronic mail to colleagues throughout the Republic of Croatia and published on preschool teachers' social networks. The aim of the research was explained and it was emphasized that participation in the research was voluntary and that trust and anonymity were granted. The questionnaire included guidelines and gave the possibility to quit at any time. The researcher could see the collected data given by the participants immediately after the questionnaire had been filled out.

### 2.4. Statistical Data Processing

The collected data were processed by the program STATISTICA 12.5 (StatSoft, Inc., Tulsa, OK, USA) and presented as charts and tables. The basic descriptive parameters (arithmetic means, standard deviation) were calculated, and Spearman's correlation coefficient and linear regression analysis were used to determine the correlation between certain variables. Statistical significance was tested at the level of $p < 0.05\%$.

## 3. Results

As illustrated, Figure 1 presents the ratio between physically active and inactive preschool teachers. The criterion was 150 min of weekly physical activity. Preschool teachers who did less were put in the category of inactive, whereas those who did more were put in the category of active. It can be observed that 24% of preschool teachers participating in the research are physically active while 76% are inactive.

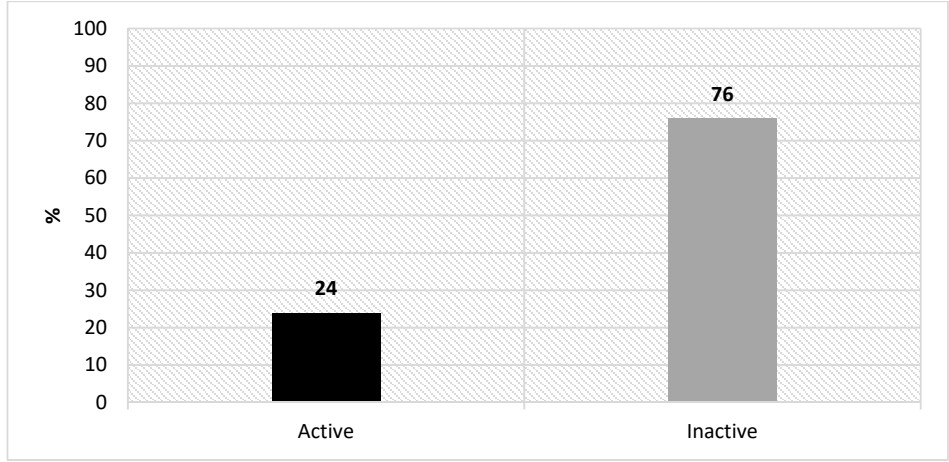

**Figure 1.** The ratio between physically active and inactive preschool teachers (%).

Figure 2 presents the average values of preschool teachers' physical activity (taking into consideration physical activities of vigorous and moderate intensity) in a time interval of one week. It is derived that preschool teachers are the least active in the category of transport (3.4 min), leisure time (28.3 min), house chores (42.5 min), and the most in the category of work (46 min).

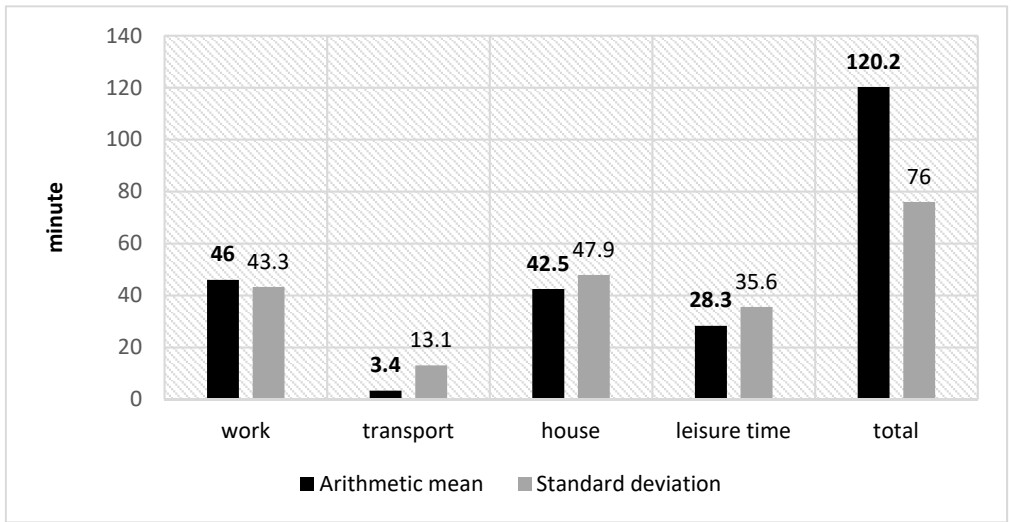

**Figure 2.** The average level of preschool teachers' physical activity of moderate and vigorous intensity is represented in minutes.

Table 1 presents the correlation between criterion variables with preschool teachers' physical activity according to domains (work, transport, house, and leisure time). A statistically significant correlation was determined among age, years of service, place of living, and physical activity; younger participants are significantly more physically active in their leisure time than their older colleagues, as are those who have fewer years of service. Participants working in a bigger city are more physically active at work, whereas those living in smaller cities are more active in doing house chores. There was no statistically significant correlation between the degree of education and physical activity.

**Table 1.** Correlation of criteria variables with preschool teachers' physical activity.

| Variables | Work | Transport | House | Leisure Time | Total |
|:---:|:---:|:---:|:---:|:---:|:---:|
| Age | −0.04 | 0.07 | 0.03 | −0.23 * | −0.10 |
| Years of service | −0.06 | 0.05 | 0.06 | −0.20 * | −0.08 |
| Place of work | 0.17 * | −0.06 | −0.16 * | −0.15 | −0.08 |
| Education | −0.07 | 0.05 | 0.09 | 0.14 | 0.09 |

* Statistical significance $p < 0.05\%$.

Table 2 presents that the preschool teacher's place of work is significantly related to their physical activity at work and during household chores. Teachers in bigger cities are significantly more physically active at work while those in smaller cities are significantly more physically active during household chores. Furthermore, the independent variables (Age, Years of service, Place of work, Education) in combination significantly affect the level of physical activity of preschool teachers in their leisure time and the overall level of physical activity. Younger preschool teachers, who live in a smaller place, with more years of work experience and a higher level of education, will be significantly more physically active in their leisure time, i.e., they will have a significantly higher overall level of physical activity.

**Table 2.** Regression summary for criteria variables with preschool teachers' physical activity.

| | | b* | Std.Err. of b* | b | Std.Err. of b | t (154) | p |
|---|---|---|---|---|---|---|---|
| Physical Activity Work R = 0.22 R2 = 0.05 F(4154) = 1.86 | Intercept | | | 37.03 | 25.97 | 1.43 | 0.16 |
| | Age | 0.09 | 0.19 | 0.36 | 0.76 | 0.47 | 0.64 |
| | Years of service | −0.19 | 0.19 | −0.80 | 0.78 | −1.02 | 0.31 |
| | Place of work | 0.18 | 0.08 | 15.85 | 7.01 | 2.26 | 0.02 |
| | Education | −0.11 | 0.08 | −5.56 | 4.28 | −1.31 | 0.21 |
| Physical Activity Transport R = 0.12 R2 = 0.014 F(4154) = 0.55 | Intercept | | | −1.52 | 7.96 | −0.19 | 0.85 |
| | Age | 0.09 | 0.19 | 0.11 | 0.23 | 0.45 | 0.65 |
| | Years of service | 0.01 | 0.19 | 0.01 | 0.24 | 0.05 | 0.96 |
| | Place of work | −0.07 | 0.08 | −1.79 | 2.15 | −0.83 | 0.41 |
| | Education | 0.08 | 0.08 | 1.19 | 1.31 | 0.91 | 0.36 |
| Physical Activity House R = 0.22 R2 = 0.05 F(4154) = 1.91 | Intercept | | | 53.80 | 28.68 | 1.88 | 0.06 |
| | Age | −0.13 | 0.19 | −0.60 | 0.84 | −0.72 | 0.47 |
| | Years of service | 0.24 | 0.19 | 1.08 | 0.86 | 1.25 | 0.21 |
| | Place of work | −0.17 | 0.08 | −16.65 | 7.75 | −2.15 | 0.03 |
| | Education | 0.12 | 0.08 | 6.94 | 4.73 | 1.47 | 0.14 |
| Physical Activity Leisure time R = 0.28 R2 = 0.08 F(4154) = 3.29 | Intercept | | | 61.57 | 20.97 | 2.94 | 0.00 |
| | Age | −0.32 | 0.18 | −1.06 | 0.61 | −1.73 | 0.09 |
| | Years of service | 0.13 | 0.19 | 0.44 | 0.63 | 0.70 | 0.48 |
| | Place of work | −0.13 | 0.08 | −9.47 | 5.66 | −1.67 | 0.10 |
| | Education | 0.08 | 0.08 | 3.53 | 3.46 | 1.02 | 0.31 |
| hysical Activity Total R = 0.15 R2 = 0.02 F(4154) = 0.84 | Intercept | | | 150.91 | 46.16 | 3.27 | 0.00 |
| | Age | −0.17 | 0.19 | −1.20 | 1.35 | −0.89 | 0.37 |
| | Years of service | 0.10 | 0.19 | 0.74 | 1.39 | 0.53 | 0.60 |
| | Place of work | −0.08 | 0.08 | −12.05 | 12.46 | −0.97 | 0.34 |
| | Education | 0.07 | 0.08 | 6.11 | 7.61 | 0.80 | 0.42 |

b*—Regression Vector; t (154)—Result of Regression Function; *p*—Value of Statistical Significance.

## 4. Discussion

The ratio between the physically active and inactive preschool teachers is worrying because 76% of preschool teachers are physically inactive, while only 24% are active. In the latest Eurostat research [7], Croatia is at the bottom regarding the frequency of engaging in physical activities. Such data are confirmed by similar research such as Heimer et al. [11], Jurakić [8], Greblo et al. [12], and by the corresponding data given by the European health survey [4]. Preschool teachers are even lower on the scale, which is highly worrying. The low level of physical activity during transport and leisure time is a particular problem. Similarly, in this research, a more negligible prevalence of the physical activity level in the transport and leisure time has also been obtained in research, including the general Croatian population [13] It is evident that there are no active individuals who are physically active in their leisure time; it is essential to emphasize that low levels of physical activity during leisure time represent the problem of inactivity the most and thus has the most decisive influence on health and life quality. Leisure time is thus seen as having a new possibility for preserving health, while poor quality of life, a consequence of physical inactivity, has an additional effect on an individual's mood, motivation, and working ability. This research conducted among early and preschool children's teachers highlights the consequence of their educational work quality. The minimal level of physical activity necessary to achieve health benefits is energy consumption in the leisure time domain of 10 MET-hour/week, 600 MET-minutes/week [14], i.e., 30 min of moderate-intensity physical activity 5 days a week. The use of only the leisure time domain as the criterion for the division of participants was done after Pedišić [15] because the valid recommendations about an adequate level of physical activity were defined, based on the research results about the health benefits of physical activity, in the domain of leisure time.

The reasons behind low values of physical activity in the category of work can be the specificities of the preschool teacher's profession which is unpredictable and flexible and does not enable doing physical activity of a certain intensity for at least 10 min in a row, which was a criterion in the questionnaire. Why are the participants insufficiently physically active in the house chores domain? The answer can be found in the house chores done by technology. For instance, robot vacuum cleaners in numerous houses have replaced vacuum cleaning. Moreover, physical activities mostly encompass works done in the garden or yard, while more and more people live in urban centres without the possibility of doing yard work.

Considering all the physical activity categories, not partially, preschool teachers are active on average for 120 min a week, which is 30 min less than the recommended minimal weekly physical activity. This information is highly worrying because preschool teachers do not have the role of a passive observer, but of an active participant in the educational process and the promoter of early behaviour patterns. They must consequently live the movement as well by the example they set and the procedures they take to become models for the acquisition of healthy life habits. It is known that the habit of engaging in physical activity is acquired from an early age [16]. Studies in recent years indicate a very low level of physical activity in young children [17] which may cause serious health problems [18]. In early and preschool-age children, processes occur in the body which are crucial for the development of potentials, so the preschool teacher is expected to be an expert who knows how to properly combine science and practice and enable each child to optimally develop these potentials [16].

An increasing number of scientists are investigating the relationship between the environment and children's health [19,20]. Physical activity is a significant human social capital and the spatial environment of cities in the world and their construction policy can significantly affect the level of physical activity of its inhabitants [21]. Apart from the global level, it is known that the spatial environment in early childhood education institutions can have a significant impact on children's physical activity [22]. The environment for a child mostly depends on the preschool teacher who needs to design it efficiently and innovatively, i.e., to make it purposeful and multifunctional [23].

In this study, the correlation between the preschool teachers' level of physical activity and chronological age is negative in the category of work and leisure time which means that as preschool teachers get older, the level of physical activity in these categories gets lower. In the category of house chores, the correlation is positive, which means that the physical activity levels are higher with older preschool teachers. This finding may be taken as specific for this research because exclusively women participated in it, and in Croatia, according to the traditional patterns of gender division, women are those who do most of the house chores [24].

The correlation of the physical activity level and years of service is also negative in the category of work and leisure time and positive in household chores which is not surprising since chronological age is almost always positively correlated to the years of service. However, a difference in the category of work has been noticed, and it can be caused by the specificity of the preschool teacher's profession where they often spend their working life in different kindergartens and are temporarily employed which causes a smaller number of years of service.

The third determining correlation is found between the variable physical activity level and the preschool teachers' place of work. A bigger city has over 15,000 citizens, a smaller one with less than 15,000 citizens. In the categories of work and transport, a higher physical activity level was determined among preschool teachers from bigger cities. This is corroborated because preschool teachers coming from bigger cities have a long way to reach their place of work than those working in smaller cities, so it is not surprising that their physical activity levels are higher in the category of transport. In house chores and leisure time, preschool teachers coming from bigger cities have a lower level of physical activity than those coming from smaller ones. The results are given by linear regression

analysis also show that preschool teachers in bigger cities are significantly more physically active at work, while those in smaller cities are significantly more physically active during household chores. In bigger cities work conditions can be much better, i.e., the spatial-material environment can be broader and more prosperous, so preschool teachers in bigger cities have much better opportunities for physical activity at work. On the other hand, preschool teachers from smaller cities often have better possibilities for physical activity in house chores because they usually live in houses with large gardens and yards that require more care from the family than preschool teachers from bigger cities. Earlier Eurobarometer survey findings have shown that people from smaller cities are more physically active than those from bigger cities, which can be linked to this category [25].

Numerous research studies show a statistically significant correlation between the level of physical activity and degree of education, as found in the research by Jurakić and Heimer [2]. However, the participants' sample must be considered, since all participants were preschool teachers. These research results did not find a statistically significant correlation between the physical activity level and degree of education. A possible reason for the lack of correlation between these variables is that the Graduate Study of Early and Preschool Education, when compared to the Undergraduate Study, has only one course linked to the kinesiology domain, and it is an elective course. Moreover, the job of a preschool teacher requires a higher degree of education than a secondary school degree, which is also clear among the participants of this research. On the other hand, when all the independent variables are analysed in combination, including the degree of education, there is a significant relationship between the level of physical activity in the leisure time domain, as well as overall. Younger preschool teachers, who live in a smaller place, with more years of work experience and a higher level of education, will be significantly more physically active in their leisure time, i.e., they will have a significantly higher overall level of physical activity.

This research is significant because it raises preschool teachers' awareness about the benefits gained from moving and empowers their kinesiological competencies for purposeful educational work. The obtained results will enable participants of the educational process a higher motivation to raise their physical activity level, which also protects their health. These research results, when compared to many research studies of the same or similar topic conducted in Europe and the world, serve as an indicator of the current situation in Croatia and are a landmark for planning future health interventions. However, it needs to be said that the questionnaire method leaves the impression of the participants' subjective opinions and although such a method may cover a larger number of participants than any other, a different methodological approach should be considered, taking the risk of including a smaller sample size. The research on this topic has shown that only a few research studies are conducted in Croatia, which includes samples of preschool teachers, so there is a space for further research that will bring social benefits. The research analysis shows areas that need interventions and motivation to gain healthy life habits and to transmit them to children of early and preschool age. Interventions are needed in every domain of physical activity in order to increase overall physical activity and thus all the benefits it brings.

## 5. Conclusions

Physical activity has been analysed in the domains of work, transport, house chores, leisure time, and all of them together per week. It was shown that almost 80% of preschool teachers do not fulfil the recommended level of physical activity, i.e., they are physically inactive. They are the most physically active at work and the least during transport and leisure time. This shows a poor life quality and jeopardized health. A statistically significant correlation was found between physical activity and age, years of service, and place of living. Younger preschool teachers and those with fewer years of service are significantly more physically active in their leisure time than older ones. Preschool teachers who work in bigger cities are more physically active at work, while those from smaller cities are

more active in household chores. There is no statistically significant correlation between education and physical activity level unless it comes to combining all the independent variables together. Then, there is a significant relationship between the level of physical activity in the leisure time domain, as well as overall.

Every person must know the consequence of insufficient physical activity leads to the risk for various human diseases and mortality. By improving one's quality of life, preschool teachers transfer the same values to early and preschool children.

**Author Contributions:** Conceptualization, V.P.; methodology, V.P. and B.F.; software, V.P.; validation, V.P.; formal analysis, V.P.; investigation, B.F.; resources, B.F.; data curation, B.F.; writing—original draft preparation, B.F.; writing—review and editing, V.P. and B.F.; visualization, B.F.; supervision, L.V.; project administration, L.V.; funding acquisition, L.V. All authors have read and agreed to the published version of the manuscript.

**Funding:** This research was funded by the University of Rijeka as a part of a project for the Establishment of a system for monitoring physical activity with modern technology in institutions of early and preschool education, under the code uniri-drustv-18-268.

**Institutional Review Board Statement:** The study was conducted in accordance with the Declaration of Helsinki and approved by the Institutional Review Board of Faculty of Teacher Education in the University of Rijeka (159. Session, 24 September 2021).

**Informed Consent Statement:** Informed consent was obtained from all subjects involved in the study.

**Data Availability Statement:** Not Applicable.

**Conflicts of Interest:** The authors declare no conflict of interest.

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
