# Peer review of "Prevalence of the Preschool Teachers’ Physical Activity Level: The Case of the Republic of Croatia"

_sustainability, doi:10.3390/su14052963_

Round 1
Reviewer 1 Report
My recommendations are as follows
Extension of the introduction. 8 bibliographic indexes on the topic of the article is far too few.
In section 2.1 also mention the standard deviation to the values presented (x ± stdv). I recommend mentioning how many of the study subjects work in large cities or in smaller cities. Also if they are from rural areas. What were the inclusion and exclusion criteria.
Figure 2, I recommend restoring, never making diagrams reporting the standard deviation.
The discussion section should focus on the results of the present study in correlation with previous studies.
This topic is studied in the literature, consequently a total of 15 bibliographic indexes are irrelevant.
Author Response
Greetings!
First of all, we would like to thank you for the review and all your suggestions, which were applied as much as possible. We apologize in advance if the corrections does not fully meet the requirements, but we were forced to follow the instructions of three reviewers and the main editor, whose suggestions did not match. The main editor suggested that we do a regression analysis and based on that the results and discussion were refined. We tried to find the best possible solution for quality in written work according to the necessary criteria. Thank you for understanding.
Kind regards,
Authors
Reviewer 2 Report
Greetings!
First, I would like to congratulate the authors for such an interesting article.
Here are my suggestions:
Line 19: Please consider rewriting the sentence into Younger preschool teachers are significantly more physically active in their free time than older ones and those with fewer years of service.
Line 23: consider rephrasing the sentence into Preschool teachers who work in a bigger city are more physically active at work, whereas those from smaller cities are more active doing their house chores
Line 39: Insufficient physical activity is becoming a significant factor of risk for developing the most common diseases of modern times ...
Line 34:...at least 150 minutes a week ...
Line 35: ...Research on this topic in the last few years is regularly conducted to detect risk factors, which could help create strategies intended to increase physical activity, especially in segments where it is the lowest [3].
Line 38:... insight into one’s health condition..
Line 40:The research analysis indicates areas that need interventions and motivation to acquire healthy everyday life habits to improve health and diminish the death rate caused by chronic non-infectious diseases [4].
Line 43: ...from the earliest age...
Line 50: ...17% exercise from three to five hours, while 28% do exercises more than five hours a week...
Line 52: 59% of the European Union citizens exercise never or rarely, while 41% do that at least once a week.
Line 55: Sweden is at the top of the list, with 70% of participants stating to do exercises at least once a week...
Line 60: In the Republic of Croatia, 56% of citizens do not do exercises or sports, while 20% do that rarely.
Line 62: According to the employment criteria, only 29.67% of male employees and 32.75% of female employees have the recommended physical activity level [8].
Line 68:... specific challenges ...
Line 69:.....evident that if they want ...
Line 72: ...for a healthy and...
Line 76: ... is a significant segment of the preschool teachers’ profession ...
Line 88: Too wordy sentence! Please rephrase it or divide it in two.
The division of the participants according to the size of their place of work was done following the criteria of the number of citizens being 15,000 (less than that, it was considered a smaller city, more than that, a bigger one).
This sentence is unclear and hard to follow:
Each question is answered in days on a scale from 0 to 7, and then in minutes from 0 to 240, where the intervals of 30 107 minutes are in the form of multiple-choice questions.
Line 109: Unclear sentence. Hard to understand. Too wordy. The sociodemographic data about the preschool teachers participating in the research, who represent the criterion variables of the research are: sex, age, years of service as a preschool teacher, place of work, and the highest achieved degree of education.
Reconsider: The sociodemographic data about the preschool teachers participating in the research represent the research's criterion variables: sex, age, years of service as a preschool teacher, place of work, and the highest achieved degree of education.
Line 114: For the variable regarding the highest degree of education, multiple-choice questions are used .....
Line 111: ...achieved the degree ..
Line 121: Data were collected in 19 days.
Wordy sentence again! The aim of the research was explained, and it was emphasized that the participation in the research was voluntary, and trust and anonymity were granted.
Line 126: The researcher could see the collected data..
STATISTICA 12.5 (StatSoft, Inc., Tulsa, Oklahoma, USA)
Line 141: ...76% are inactive. ...
Line 172: ...which is highly worrying. ...
Line 173: ....free time is a particular problem....
Line 173: Similarly to this research, a more negligible prevalence of the physical activity level in the transport and free time has also been obtained in research, including the general Croatian population [13]. ...
Line 177: ...it is essential to emphasize t...
Line 178: ...thus the most decisive influence on health ...
Line 179: Free time is thus seen as a new possibility of preserving health, while poor quality of life, a consequence of physical inactivity, has an additional effect on an individual’s mood, motivation, and working ability.
Line 183: The minimal level of physical activity necessary to achieve health benefits is the energy consumption in the free time domain of 10 MET-hour/week, 600 MET-minutes/week [14], i.e. 30 minutes of moderate-intensity physical activity five days a week. ...
Line 194: Why are the participants insufficiently physically active in the house chores domain?
Line 195: The answer can be found in the house chores done by technology. For instance, robot vacuum cleaners in numerous houses have replaced vacuum cleaning.
Line 203: ...is highly worrying because ...
Line 207: It is known that the habit of engaging in physical ....
Line 208:....from the earliest age...
Line 211: ..which means that as preschool teachers ...
On the other hand, in the category of house chores, the correlation is positive, which means.
Line 214: ...women always do most of the hose chores [17].
Line 217: The correlation of the physical activity level and years of service is also negative in the category of work and free time and positive in house chores, which is not surprising since chronological age is almost always positively correlated to the years of service.
Line 224: The third determining correlation....
Line 222_...where they often spend ...
Line 198: ...activities mainly encompass ...
Line 226: In the categories of work and transport, a higher physical activity level was determined among preschool teachers from bigger cities. In house chores and free time, preschool teachers coming from bigger cities have a lower level of physical activity than those coming from smaller ones.
Line 231: ...have a long way to reach ....
Line 234: ---environment can be broader and more prosperous,--
Line 235: On the other hand, preschool teachers from smaller cities often have better possibilities for physical activity in house chores because they usually live in houses with large gardens and yards that require more care for the family than preschool teachers from bigger cities.
Line 240: ...which can be linked to...
Line 243: ...as found in the research by Jurakić ...
However, the participants' sample must be considered since all participants were preschool teachers.
Line 253: This research is significant in that it raises preschool teachers' awareness about the benefits gained from moving and empowers their kinesiological competencies for purposeful educational work.
Line 255: The obtained results will enable participants of the educational process a higher motivation to raise their physical activity level, which also protects their health.
Line 257: These research results are ...Europe and the world and serve as an indicator of the current situation in Croatia and a landmark for planning future health interventions.
Line 262: The research on this topic has shown that only a few research studies are conducted in Croatia, which includes samples of preschool teachers, so there is a space for further research that will bring social benefits.
Line 270:
A statistically significant correlation between physical activity and age, years of service and place of living. (This appears to be a sentence fragment. Consider rewriting it as a complete sentence.)
Line 273: ...with fewer years of service ...
Line 278: Every person has to be aware of the consequences of insufficient physical activity, leading to risk for various human diseases and mortality..
Line 280: By improving one’s quality of life...
You should state how findings could help improve the practice and theory.
The application area has to be sustained in more detail.
Line 110: Please use "gender" instead of "sex".
Did you use the snowboard method in forwarding the questionnaire?
House chores (42.5 minutes) could be explained with the participants' gender?
In Table 1 should the "Place of work" be replaced with the Place of physical activity?
Please check the space between words in the article. There are many technical mistakes. The delineation of strengths and weaknesses of the methodology has to be emphasised.
Overall, it is excellent work.
Wish you all the best in your future work!
Author Response
Greetings!
First of all, we would like to thank you for the comprehensive review and all your suggestions, which were applied in the mentioned lines. We apologize in advance if the corrections does not fully meet the requirements, but we were forced to follow the instructions of three reviewers and the main editor, whose suggestions did not match. The main editor suggested that we do a regression analysis and based on that the results and discussion were refined. We tried to find the best possible solution for quality in written work according to the necessary criteria. Thank you for understanding.
Kind regards,
Authors
Reviewer 3 Report
Review comments can be found in the attached file.

Author Response

(The authors gave the same response as above.)

Round 2
Reviewer 1 Report
no comments